# Electrochemical cobalt-catalyzed semi-deuteration of alkynes to access deuterated Z-alkenes

Wen-Jie Feng[1,2], Zhe Chang[1,2], Xi Lu [1,3] ✉ & Yao Fu [1,3] ✉

Deuterium labeling has found extensive applications across various research fields, including organic synthesis, drug design, and molecular imaging. Electrocatalytic semi-hydrogenation of alkynes offers a viable route for the synthesis of Z-alkenes, yet it falls short in achieving the semi-deuteration of these compounds. In this study, we report an electrochemical cobalt-catalyzed transfer deuteration reaction that proficiently accomplishes the semi-deuteration of alkynes, yielding Z-configuration deuterated alkene products. This reaction utilizes cost-effective cobalt salts as catalysts and employs $D_2O$ and AcOD (acetic acid-d) as economical and efficient deuterium sources, underscoring its practicality and feasibility. The reaction demonstrates a broad alkyne substrate scope, high reaction efficiency, good functional group compatibility, excellent Z-selectivity, and a remarkable degree of deuteration rate.

Since the first discovery of the deuterium element in 1932, deuterium labeling has found extensive applications across various research fields, including organic synthesis, drug design, and molecular imaging[1–10]. In particular, it plays an important role in drug metabolism studies, where the precise incorporation of deuterium atoms into molecular structures aids researchers in tracking the metabolic pathways of drugs within biological systems, thereby enhancing the precision and efficacy of drug design. The higher bond dissociation energy of C-D compared to C-H bonds renders deuterated compounds generally more metabolically stable in vivo than their hydrogenated counterparts. Substituting hydrogen atoms with deuterium can improve drug metabolic stability, prolong pharmacological effects, and reduce adverse reactions.

On the other hand, alkenes, characterized by their carbon-carbon double bonds, exhibit unique reactivity and versatility. As shown in Fig. 1a, numerous natural products and pharmaceutical molecules contain Z-olefinic structures[9]. However, Z-alkenes are thermodynamically less stable than E-alkenes, making the efficient and highly selective synthesis of Z-alkenes a crucial research direction in organic chemistry[11–18]. For instance, metal-catalyzed systems, exemplified by Lindlar catalysts[19–25], utilize hydrogen atmospheres to reduce alkynes,

achieving the semi-hydrogenation of alkynes into Z-alkenes[26–32]. Analogously, the semi-deuteration of alkynes offers a convenient route to deuterated Z-alkenes[33–36]. Nevertheless, this reaction faces several challenges. Firstly, the resulting deuterated alkene products may readily undergo further reduction under the same catalytic conditions, generating deuterated alkane by-products[37–41]. Secondly, the semi-deuteration process may yield a mixture of E- and Z-alkene stereoisomers, posing difficulties in the effective separation of Z-alkenes due to their similar polarities and sizes[31,42,43]. Lastly, to meet the demands of drug development and mechanistic studies involving deuterium substitution[44], high deuteration rates are essential for alkyne semi-deuteration reactions.

Electrochemistry has proven to be an excellent platform for developing cost-effective and sustainable organic reactions[45–71]. Electrocatalytic semi-hydrogenation of alkynes provides a method for preparing Z-alkenes (Fig. 1b). Baran and co-workers achieved excellent work of Z-selective semi-hydrogenation of alkynes through a cobalt-electrocatalytic hydrogen-atom transfer process[72]. Kaeffer and co-workers utilized [Ni(bpy)$_3$]$^{2+}$ as a catalyst for the electrocatalytic Z-selective semi-hydrogenation of alkynes[73]. In addition, Kaeffer and co-workers also demonstrated the potential of BzOD (benzoic acid-d) for

[1]State Key Laboratory of Precision and Intelligent Chemistry, University of Science and Technology of China, Hefei, China. [2]These authors contributed equally: Wen-Jie Feng, Zhe Chang. [3]These authors jointly supervised this work: Xi Lu, Yao Fu. ✉e-mail: luxi@mail.ustc.edu.cn; fuyao@ustc.edu.cn

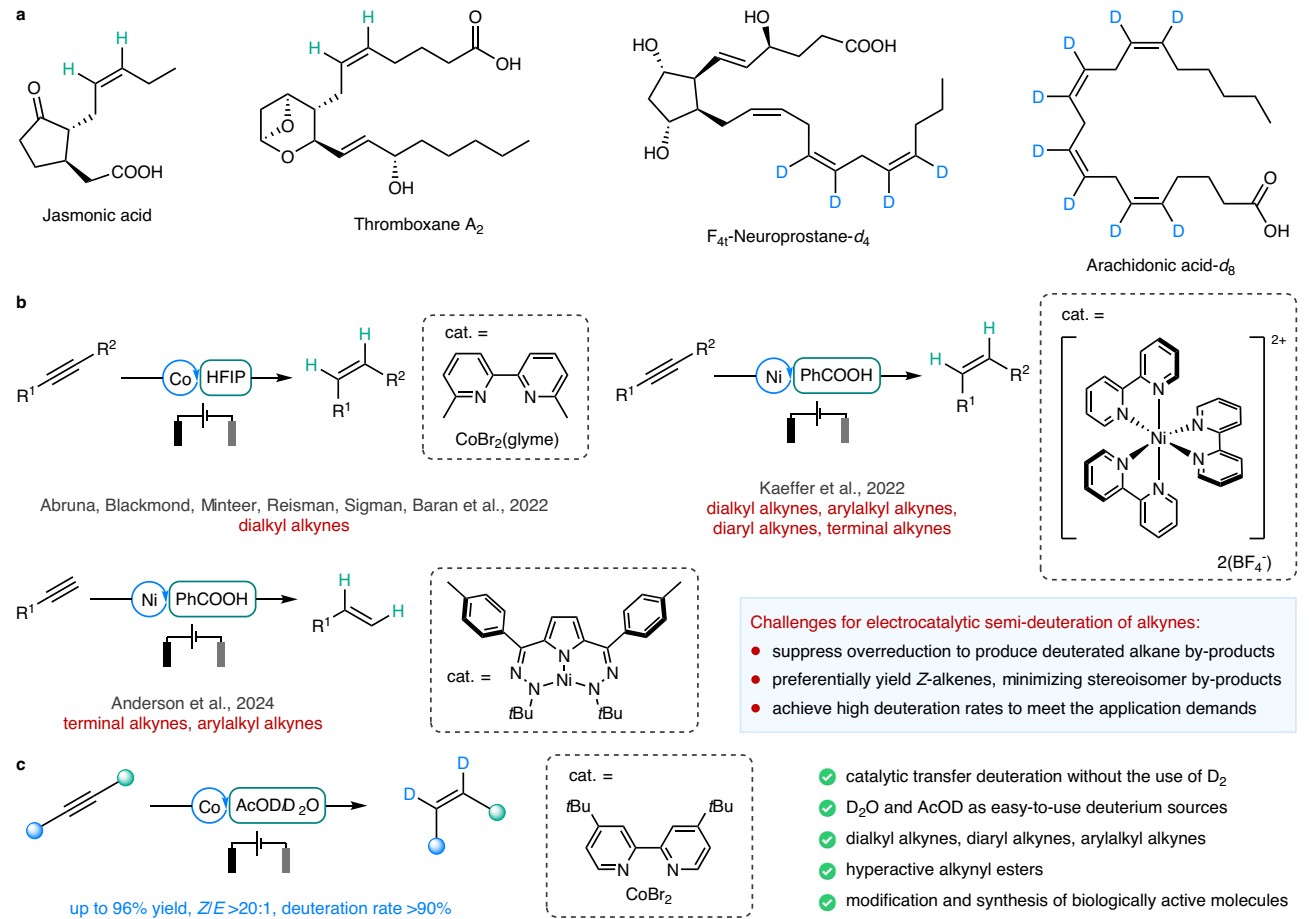

**Fig. 1 | Electrocatalytic semi-hydrogenation and semi-deuteration of alkynes.**
**a** *Z*-alkenes and deuterated alkenes have extensive applications in organic synthesis, drug design, and molecular imaging. **b** Convenient synthesis of *Z*-alkenes can be accomplished via electrocatalytic semi-hydrogenation of alkynes. **c** This work: electrochemical cobalt-catalyzed semi-deuteration of alkynes to access deuterated *Z*-alkenes. cat. catalyst, HFIP 1,1,1,3,3,3-hexafluoro-2-propanol, glyme 1,2-dimethoxyethane, AcOD acetic acid-*d*.

achieving electrocatalytic semi-deuteration of alkynes[32]. Anderson and co-workers realized the electrocatalytic *Z*-selective semi-hydrogenation of terminal alkynes with a dihydrazonopyrrole Ni complex via a ligand-based hydrogen-atom transfer pathway[74]. Despite achievements that have been made, these electrochemical reduction systems are difficult for realizing the semi-deuteration of alkynes. Therefore, there is an urgent need to develop an electrochemical reduction system for achieving semi-deuteration of alkynes. We envision developing an electrochemical transfer deuteration reaction involving in situ generation of active deuterium species to achieve the semi-deuteration of alkynes. This approach offers a safe, sustainable, and environmentally friendly alternative to conventional alkyne semi-deuteration reactions.

Herein, we report an electrochemical cobalt-catalyzed transfer deuteration reaction that enables the semi-deuteration of alkynes to synthesize *Z*-configuration deuterated alkene products (Fig. 1c). This reaction employs inexpensive cobalt salts as catalysts and harnesses $D_2O$ and AcOD as economical and efficient deuterium sources. This reaction demonstrates a broad alkyne substrate scope, high reaction efficiency, good functional group compatibility, excellent *Z*-selectivity, and a remarkable degree of deuteration rate.

## Results
### Reaction development
We selected alkyne (**1**) as the model substrate to screen the optimal conditions for the semi-deuteration reaction (Fig. 2a). Through the screening of reaction conditions, we found that the combination of

$CoBr_2$ and dtbpy as catalysts, AcOD and $D_2O$ as deuterium sources, $PPh_3$ as a sacrificial reductant, $TBABF_4$ as an electrolyte, C(+)/C(-) as electrodes, using a mixed solvent of DMAc and $D_2O$, and electrolysis at a constant current of 2.5 mA, could smoothly convert alkyne substrate **1** to deuterated alkene product **2** with a GC yield of 92%, an isolated yield of 88%, a *Z/E* selectivity ratio of >20:1, and deuteration rates of 92% and 91%, respectively (entry 1). We compared the influence of different variables on the reaction effect and categorized them for presentation. We discovered that several cobalt salts, including $Co(OAc)_2$, $Co(acac)_2$, $CoCl_2$, could replace $CoBr_2$ as the reaction catalyst, exhibiting good catalytic performance (entries 2–4). However, when Co-salen (cobalt *t*Bu,*t*Bu-cyclohexylsalen) was used instead of the combination of $CoBr_2$ and dtbpy, the alkyne conversion rate significantly decreased, resulting in reduced reaction efficiency (entry 5). For different bidentate nitrogen ligands, the results showed that using ligand **L1**, which has a similar structure to dtbpy, could achieve generally satisfactory reaction yields (entry 6); whereas using ligand **L2**, with methyl substitution adjacent to the nitrogen atom, significantly reduced the reaction conversion and yield (entry 7). We also screened $Et_3N$, TEOA, DIPEA, and DABCO, as alternative sacrificial reagents (entries 8-11). However, their yields or deuteration rates are far inferior to those of $PPh_3$. Using DMF or DMSO as the solvent yielded results comparable to those obtained with DMAc as the solvent (entries 12-13), however, a much lower yield was obtained in MeCN (entry 14). If some of the high-priced reagents can be replaced with non-deuterated alternatives, especially without the need for large amounts of $D_2O$, it would be a great help for our reaction to move towards industrial-scale

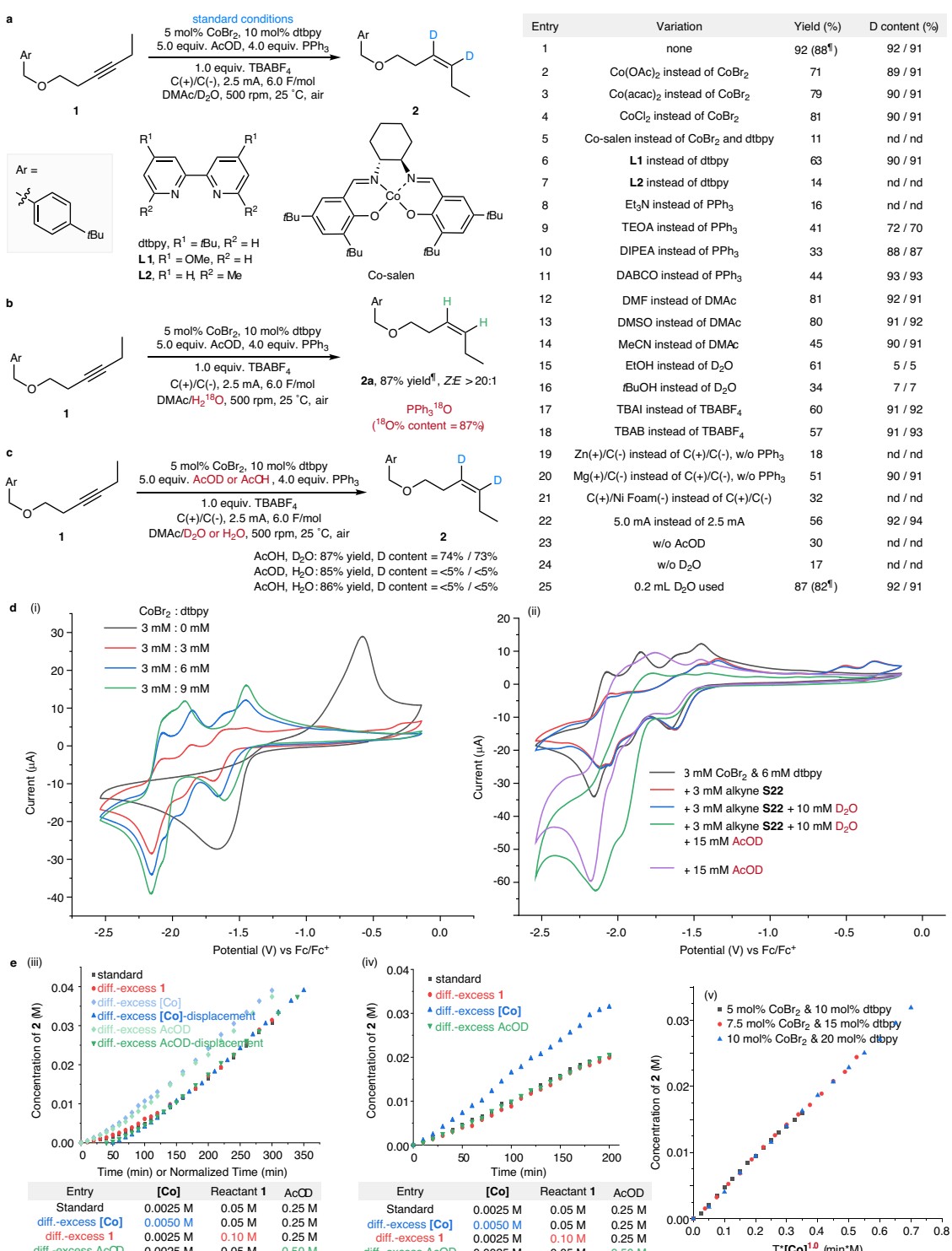

| Entry | Variation | Yield (%) | D content (%) |
|---|---|---|---|
| 1 | none | 92 (88¶) | 92 / 91 |
| 2 | Co(OAc)₂ instead of CoBr₂ | 71 | 89 / 91 |
| 3 | Co(acac)₂ instead of CoBr₂ | 79 | 90 / 91 |
| 4 | CoCl₂ instead of CoBr₂ | 81 | 90 / 91 |
| 5 | Co-salen instead of CoBr₂ and dtbpy | 11 | nd / nd |
| 6 | L1 instead of dtbpy | 63 | 90 / 91 |
| 7 | L2 instead of dtbpy | 14 | nd / nd |
| 8 | Et₃N instead of PPh₃ | 16 | nd / nd |
| 9 | TEOA instead of PPh₃ | 41 | 72 / 70 |
| 10 | DIPEA instead of PPh₃ | 33 | 88 / 87 |
| 11 | DABCO instead of PPh₃ | 44 | 93 / 93 |
| 12 | DMF instead of DMAc | 81 | 92 / 91 |
| 13 | DMSO instead of DMAc | 80 | 91 / 92 |
| 14 | MeCN instead of DMAc | 45 | 90 / 91 |
| 15 | EtOH instead of D₂O | 61 | 5 / 5 |
| 16 | tBuOH instead of D₂O | 34 | 7 / 7 |
| 17 | TBAI instead of TBABF₄ | 60 | 91 / 92 |
| 18 | TBAB instead of TBABF₄ | 57 | 91 / 93 |
| 19 | Zn(+)/C(-) instead of C(+)/C(-), w/o PPh₃ | 18 | nd / nd |
| 20 | Mg(+)/C(-) instead of C(+)/C(-), w/o PPh₃ | 51 | 90 / 91 |
| 21 | C(+)/Ni Foam(-) instead of C(+)/C(-) | 32 | nd / nd |
| 22 | 5.0 mA instead of 2.5 mA | 56 | 92 / 94 |
| 23 | w/o AcOD | 30 | nd / nd |
| 24 | w/o D₂O | 17 | nd / nd |
| 25 | 0.2 mL D₂O used | 87 (82¶) | 92 / 91 |

**Fig. 2 | Reaction development. a** Optimization of reaction conditions. **b** Isotope labeling experiments. **c** Comparison of deuterium sources. **d** Cyclic voltammetry experiments. (i) Comparison of cyclic voltammograms obtained from different ratios of cobalt salts and ligands. (ii) Cyclic voltammogram of cathode reaction system. **e** Reaction kinetic experiments. (iii) Different-excess experiments under 2.5 mA current. (iv) Different-excess experiments under 7.5 mA current. (v) Burés graphical rate analysis for cobalt-salt and ligand complex under 7.5 mA current. Standard conditions: **1** (1.0 equiv., 0.050 mol/L), CoBr₂ (5 mol%, 0.0025 mol/L), dtbpy (10 mol%, 0.005 mol/L), AcOD (5.0 equiv., 0.250 mol/L), PPh₃ (4.0 equiv., 0.200 mol/L), TBABF₄ (1.0 equiv., 0.050 mol/L), C(+)/C(-), 2.5 mA, 6.0 F/mol, DMAc

(3.6 mL), D₂O (0.40 mL), 500 rpm, 25 °C, under air. 0.2 mmol scales. GC (gas chromatography) yield. Deuterium content was determined by ¹H-NMR (nuclear magnetic resonance) spectroscopy after isolation. ¶Isolated yields are given in parentheses. Co(OAc)₂ cobalt acetate; Co(acac)₂ bis(acetylacetonato)cobalt; DMAc N,N-dimethylacetamide, DMF N,N-dimethylformamide, DMSO dimethyl sulfoxide, TBABF₄ tetrabutylammonium tetrafluoroborate, TBAI tetrabutylammonium iodide, TBAB tetrabutylammonium bromide, TEOA triethanolamine, DIPEA N,N-diisopropylethylamine, DABCO triethylenediamine, tBu tert-butyl, rpm revolutions per minute, nd not determined, D content was not detected when the GC yield was <30%, AcOH acetic acid, **S22**, hex-1-yn-1-ylbenzene.

production. However, using AcOD in combination with mixed solvents such as DMAc/EtOH or DMAc/tBuOH yielded products with unsatisfactory deuteration rates (entries 15-16). Using TBAI or TBAB as the electrolyte instead of TBABF$_4$ resulted in moderate yields (entries 17-18). We utilized metals as sacrificial anodes to replace the use of PPh$_3$ reductants. Zn(+) and Mg(+) electrodes were used to replace the graphite electrode as the anode, while the cathode remained the graphite electrode (entries 19-20). It was found that the Mg(+) electrode performed better than Zn(+) electrode. Replacing the graphite cathode electrode with a Ni Foam(-) electrode, while keeping the anode as the graphite electrode, yielded unsatisfactory results (entry 21). When we doubled the reaction current to 5.0 mA for constant-current electrolysis, the reaction yield decreased significantly (entry 22). Control experiments demonstrated that the combination of AcOD and D$_2$O was crucial for achieving satisfactory reaction yields, which may be related to modulating the acidity of the reaction system (entries 23-24). To minimize the use of costly deuterated solvents, we tried halving the amount of D$_2$O used. The deuteration rate of the reaction remained almost unaffected, but there was a slight decrease in yield (entry 25). Finally, due to the minimal changes in variables such as catalysts, ligands, deuterium sources, electrolytes, electrodes, sacrificial reagents, and current, they have a negligible influence on the Z/E-stereoselectivity.

We conducted isotope labeling experiments using heavy-oxygen water (H$_2$$^{18}$O) instead of D$_2$O (Fig. 2b). High-resolution mass spectrometry (HRMS) results confirmed the formation of PPh$_3$$^{18}$O and indicated that the $^{18}$O atoms originated from H$_2$$^{18}$O. We conducted a comparison among different combinations of deuterium sources (Fig. 2c). Various permutations and combinations of H$_2$O, D$_2$O, AcOH, and AcOD were examined. We observed that the reaction yield was minimally affected, whereas the deuteration rate was significantly influenced. When the combination of D$_2$O and AcOH was used, the deuteration rates dropped to 74% and 73%. Conversely, with the combination of H$_2$O and AcOD, the introduction of many protons in the solvent resulted in products containing almost no deuterium atoms. These comparative experiments demonstrated that significant H/D exchange occurred in the reaction system.

Cyclic voltammetry experiments were conducted to provide more mechanistic information (Fig. 2d). We examined various combinations of cobalt-salt and ligand at different ratios (Fig. 2d, i). In the absence of a ligand dtbpy, CoBr$_2$ exhibits a single reduction peak at −1.67 V. Upon the addition of ligand dtbpy, the potential of this reduction peak shifts to −1.61 V, presumably corresponding to the reduction potential of Co$^{II}$/Co$^{I}$, and a new reduction peak emerges at -2.16 V, which we speculate is the reduction peak of Co$^{I}$/Co$^{0}$. During the gradual increase of ligand dtbpy, some shoulder peaks emerge and then disappear, demonstrating that multiple species of the cobalt catalyst exist when cobalt-dtbpy coordination is unsaturated, but this does not affect the main reduction peaks. Then, we conducted the cyclic voltammogram of the cathode reaction system (Fig. 2d, ii). Upon the addition of alkynes (red line) or alkynes and D$_2$O (blue line) to the cobalt-ligand complex, the peak current at −1.61 V has no significant changes, and the peak current at −2.16 V decreases, suggesting that coordination may occur between the alkyne and cobalt species to affect the Co$^{I}$ reduction. When AcOD (purple line) is directly added or alkynes, D$_2$O, and AcOD (green line) are simultaneously added to the cobalt-ligand complex, the peak current at −1.61 V decreases, and the reduction peak around −2.16 V increases significantly, suggesting new cobalt species generated, may be the formation of CoD species[71,72]. Based on these results, we propose a possible reaction mechanism involving CoD species, which originate from the deuteration of low-valent cobalt species. After the insertion of the CoD species into the alkyne, it undergoes a deuteration step to produce the semi-deuteration products. Moreover, cyclic voltammetry experiments indicate that the system relies on the more acidic AcOD.

Finally, we conducted reaction kinetic experiments to aid in understanding the reaction mechanism (Fig. 2e). At 2.5 mA, the reaction rates of the alkyne, cobalt catalyst, and AcOD under different-excess conditions are basically the same. The initial induction period rates for the excess cobalt catalyst and AcOD are significantly faster than those of the standard reaction. However, by shifting the time axis, it can be observed that the mid-reaction rates are essentially consistent, all exhibiting zero-order kinetic behavior (Fig. 2e, iii). At 7.5 mA, the reaction rates remain constant for different excesses of alkyne and AcOD, and a significant increase in reaction rate is observed upon increasing the amount of cobalt catalyst, demonstrating that the kinetic order of both alkyne and AcOD remains zero (Fig. 2e, iv). Through Burés graphical analysis of the reaction rates at different catalyst concentrations, it can be determined that the reaction rate has a first-order kinetics with respect to the cobalt catalyst at 7.5 mA (Fig. 2e, v). The above results indicate that the rate-determining step of the reaction depends on the matching relationship between the current magnitude and the catalyst loading.

## Substrate scope

Under optimal reaction conditions, we conducted an extensive investigation into the substrate scope of the alkyne semi-deuteration reaction (Fig. 3). This reaction demonstrates broad applicability towards various types of alkyne substrates, encompassing dialkyl alkynes (**2-13**), diaryl alkynes (**14-21**), arylalkyl alkynes (**22-25**), and substituted alkynyl esters (**26-27**), all of which afford the desired products with moderate to excellent yields. For these di-substituted alkynes, the reaction exhibits remarkable semi-deuteration selectivity, yielding predominantly Z-deuterated alkene products with negligible over-deuterated alkane by-products detected. Furthermore, the reaction boasts excellent stereoselectivity, with Z/E selectivity ratios consistently exceeding 20:1 and deuterium incorporation rates generally maintaining above 90%. This high degree of deuterium incorporation facilitates research on the activity of deuterated drugs, the design and development of deuterated functional materials, as well as the synthesis of high-deuterium-enriched raw materials for deuteration experiments. In the case of diaryl alkynes (**14-21**), the electronic effects of the aromatic rings do not significantly impact the reaction's selectivity or deuterium incorporation rate.

To validate the functional group compatibility, we examined the catalytic transfer deuteration reactions using derivatives of physiologically active molecules. The results indicate that a diverse array of bioactive molecule derivatives, including flurbiprofen (**9**), oxaprozin (**10**), niflumic acid (**11**), probenecid (**12**), serine (**13**), and borneol (**27**), can all successfully participate in the reaction, yielding semi-deuterated products. Throughout this process, functional groups such as esters (**3-4**), silyl ethers (**5**), fluorine (**9**), trifluoromethyl (**11**), sulfonamides (**12**), and secondary amides bearing N-H bonds (**13**) do not significantly interfere with the reaction. Additionally, the reaction is compatible with a wide range of heterocyclic frameworks, including thiophene (**6**), furan (**7**), oxazole (**10**), and pyridine (**11**) rings, further expanding the scope of its applicability.

## Synthetic applications

We further investigated the extended applications of this alkyne semi-deuteration reaction (Fig. 4). By adjusting the reaction conditions (Fig. 4a), specifically substituting AcOH for AcOD and H$_2$O for D$_2$O, we have successfully achieved the semi-hydrogenation of internal alkynes (**28-29**) and terminal alkynes (**30**), with the semi-hydrogenation of internal alkynes exhibiting remarkable stereoselectivity (Z:E > 20:1). The alkenyl double bond, an indispensable structural unit in drug molecules and natural products, underscores its paramount importance. In the direction of drug metabolism research and deuterated drug development, deuterium labeling technology holds an irreplaceable position. Consequently, we applied the semi-deuteration

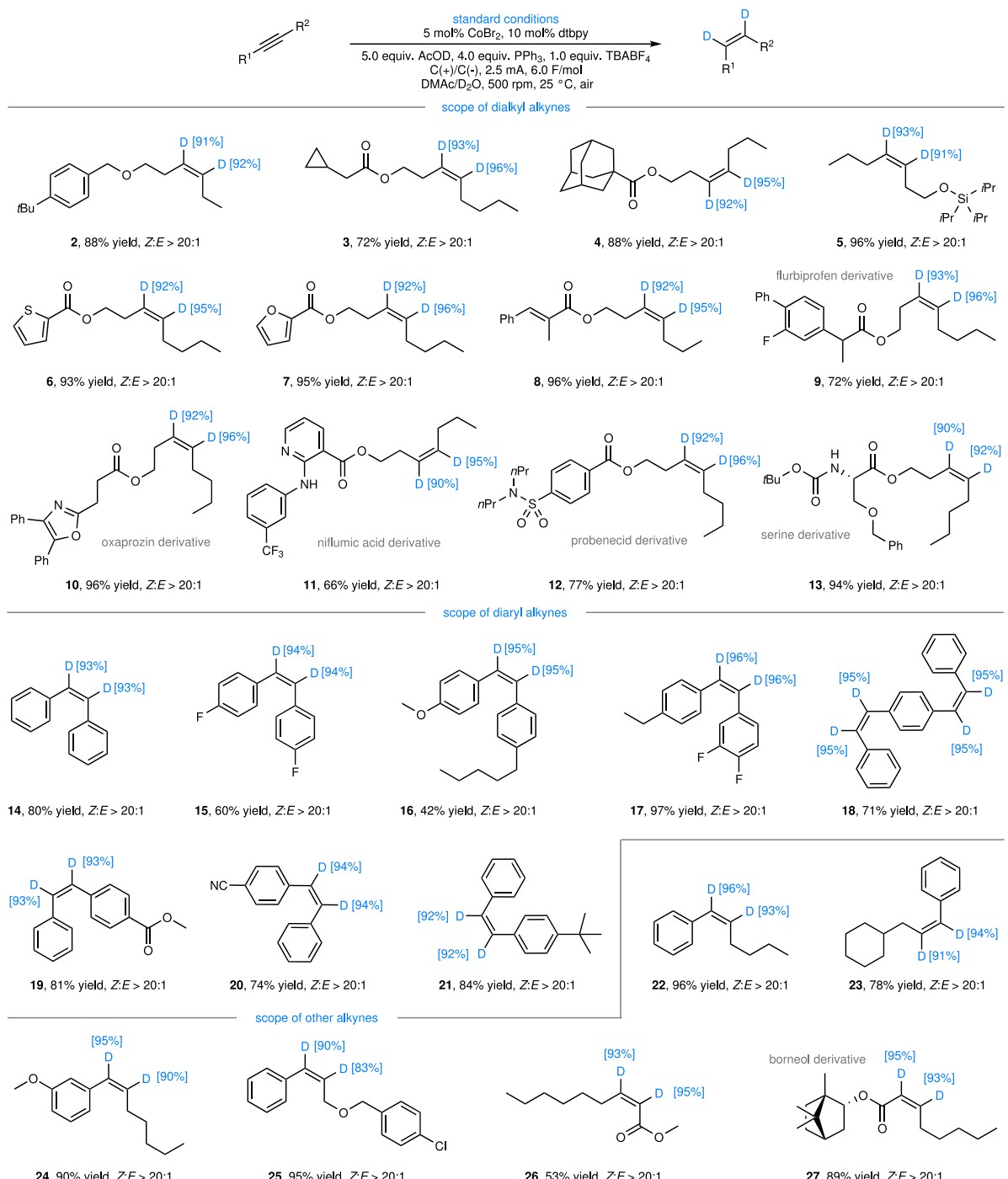

**Fig. 3 | Substrate scope of the alkyne semi-deuteration reaction.** Standard conditions: alkyne (1.0 equiv., 0.050 mol/L), CoBr$_2$ (5 mol%, 0.0025 mol/L), dtbpy (10 mol%, 0.005 mol/L), AcOD (5.0 equiv., 0.250 mol/L), PPh$_3$ (4.0 equiv., 0.200 mol/L), TBABF$_4$ (1.0 equiv., 0.050 mol/L), C( + )/C(-), 2.5 mA, 6.0 F/mol, DMAc (3.6 mL), D$_2$O (0.40 mL), 500 rpm, 25 °C, under air. 0.2 mmol scales, isolated yield. $i$Pr isopropyl, $n$Pr $n$-propyl.

reaction to the alkyne precursors of drug molecules and natural products, aiming to synthesize their deuterium-labeled analogs (Fig. 4b). Exemplified by the cerebrovascular disease treatment drug stugeron (**32**), the antifungal dermatological drug naftifine (**34**), and the active ingredient capsaicin (**36**) from peppers, we have successfully synthesized the $Z$-configuration deuterium-labeled analogs of these molecules. This electrochemical cobalt-catalyzed reaction has simplicity in operation and scalability to gram-scale, demonstrating its potential for practical applications (Fig. 4c). Through this reaction, we have smoothly produced 0.941 g of alkyne semi-deuteration product while

maintaining a high isolated yield of 85%, which convincingly demonstrates that the reaction excels not only at small scales but also retains its efficiency and good controllability in larger-scale production. However, there remains a substantial gap between our gram-scale reaction and industrial-scale production. In particular, industrial-scale production necessitates careful consideration of core issues such as the recycling of D$_2$O and the waste treatment. Furthermore, we achieved the efficient transformation of deuterated alkene products into diverse target deuterated products by using strategies such as epoxidation, Pd-catalyzed remote dioxygenation, and Ni-catalyzed

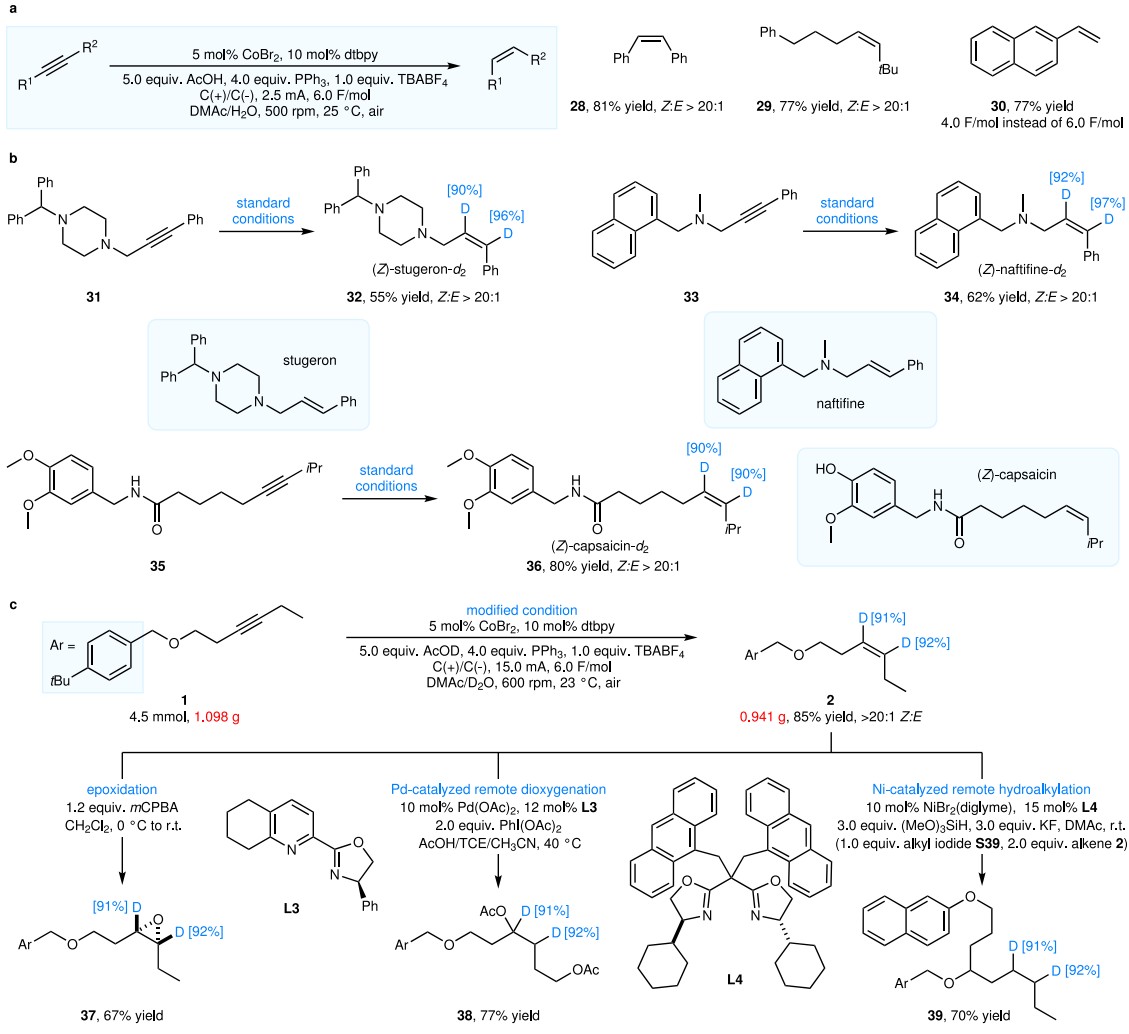

**Fig. 4 | Application expansion. a** Alkyne semi-hydrogenation. **b** Synthesis of deuterated analogs of bioactive molecules. **c** Gram-scale reaction and subsequent conversion. Standard conditions as shown in Fig. 3. For Fig. 4a and Fig. 4c, modified conditions were used. 0.2 mmol scales, isolated yields, unless otherwise noted. For compounds **38** and **39**, the absolute positions of the deuterium atoms cannot be completely confirmed. *m*CPBA, 3-chloroperoxybenzoic acid; r.t. room temperature, Ac acetyl, TCE tetrachloroethane, diglyme 2-methoxyethyl ether, **S39** 2-(3-iodopropoxy)naphthalene.

remote hydroalkylation. This series of accomplishments fully showcases the efficiency of this catalytic system in constructing complex molecular structures.

In conclusion, we have developed an electrochemical cobalt-catalyzed transfer deuteration reaction, which enables the semi-deuteration of alkynes to synthesize *Z*-configuration deuterated alkene products. This reaction employs inexpensive cobalt salts as catalysts and utilizes D$_2$O and AcOD as efficient deuterium sources. The reaction demonstrates broad applicability to various types of alkyne substrates, including dialkyl alkynes, diaryl alkynes, arylalkyl alkynes, and substituted alkynyl esters. It features high reaction efficiency, good functional group compatibility, and excellent stereoselectivity, with a *Z/E*-ratio generally exceeding 20:1 and a deuteration rate consistently maintained above 90%. This reaction is suitable for semi-deuteration modification of biologically active molecular derivatives and can also be applied to the synthesis of deuterium-labeled analogs of pharmaceutical molecules and natural products, showcasing its efficiency in constructing complex molecular structures.

## Methods
**Electrochemical cobalt-catalyzed semi-deuteration of alkynes.** To a 5 mL vial equipped with a magnetic stirring bar, CoBr$_2$ (0.01 mmol, 5 mol%), dtbpy (0.02 mmol, 10 mol%), TBABF$_4$ (0.20 mmol, 1.0 equiv.),

and PPh$_3$ (0.80 mmol, 4.0 equiv.) were added. Subsequently, DMAc (3.6 mL) was added, and the solution was stirred for 5 minutes to ensure complete dissolution. To this solution, Na$_2$SO$_4$ was added for the purposes of drying and dehydration. After allowing the solution to stand for 30 minutes, Na$_2$SO$_4$ was removed by filtration, and the resulting solution was then transferred into a 5 mL ElectraSyn vial that was equipped with a magnetic stirring bar. Another method involves preparing 30 mL of a standard solution in a 50 mL vial, which contains CoBr$_2$ (2.78 mmol/L), dtbpy (5.56 mmol/L), PPh$_3$ (222 mmol/L), TBABF$_4$ (55.6 mmol/L) in DMAc. This solution was dried with Na$_2$SO$_4$, and subsequently, the Na$_2$SO$_4$ was removed by filtration. Afterward, 3.6 mL of this standard solution was transferred into a 5 mL ElectraSyn vial equipped with a magnetic stirring bar. The standard solution is sufficient for use in a batch of 6 parallel reactions. Following the solution preparation method, alkyne (0.20 mmol, 1.0 equiv.), AcOD (1.0 mmol, 5.0 equiv.), and D$_2$O (0.40 mL), were added to the solution in ElectraSyn vial via syringe. The ElectraSyn vial cap, equipped with an anode (graphite) and a cathode (graphite), was inserted into the reaction mixture. After pre-stirring for 5 minutes, the reaction mixture was electrolyzed under a constant current of 2.5 mA for 6 F/mol. The voltage of the electrolytic cell was approximately 2.0 V and increased over time. After the reaction, the electrodes were washed with ethyl acetate, and the organic phases were collected. The mixture was diluted with

H₂O, followed by extraction with ethyl acetate, dried with anhydrous Na₂SO₄, and concentrated in vacuo. The residue was purified by flash column chromatography to yield the target product.

## Data availability

Data generated in this study are provided in the main text and Supplementary Information files. The general information, optimization of reaction conditions, experimental procedures, and characterization of all new compounds are provided in the Supplementary Information files. Data supporting the findings of this manuscript are also available from the corresponding author upon request.

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

## Acknowledgements

Financial support was received from the National Natural Science Foundation of China (22293011 for Y.F., T2341001 for Y.F., 22371273 for X.L.), the Youth Innovation Promotion Association of the Chinese Academy of Sciences (2023476 for X.L.), and the Natural Science Foundation of Anhui Province (2208085J26 for X.L.). This work was partially carried out at the Instruments Center for Physical Science of University of Science and Technology of China. The authors acknowledge Jie Yang (USTC) for the discussion on industrial-scale production.

## Author contributions

X.L. and Y.F. directed the project and conceived the idea. W.J.F. and Z.C. designed and performed the experiments. X.L. wrote the manuscript draft with the help of Z.C. All of the authors participated in the discussion and preparation of the manuscript.

## Competing interests

The authors have applied for a patent that involves personal financial interests. All authors are not on the patent, and all authors on the patent don't have any additional financial or non-financial competing interests. Patent title: A Novel Method for the Synthesis of Deuterated Alkenes through Electrochemical Reduction of Alkynes. Patent applicant: University of Science and Technology of China. Names of inventors: Yao Fu, Wen-Jie Feng, Zhe Chang, Xi Lu, and Jing Shi. Application number: 202411088344.2. Status of application: the invention patent application has entered the substantive examination stage. Specific aspect of the manuscript covered in the patent application: the principle and partial conditions for the synthesis of deuterated alkenes through electrochemical reduction of alkynes.
