## [Transparent Peer Review file · Nature Communications]

Electrochemical Cobalt-Catalysed Semi-Deuteration of Alkynes to Access Deuterated Z-Alkenes

Corresponding Author: Professor Xi Lu

Version 0:

Reviewer comments:

Reviewer #1

(Remarks to the Author)

In this study, the authors report the efficient synthesis of deuterated alkenes through electrochemical reduction using inexpensive deuterated sources, namely deuterated water and deuterated acetic acid. Compared to previous literature, this work is highly novel and advantageous in several aspects as the following: 1) the reaction effectively inhibits over-reduction, thereby minimizing the generation of alkane by-products; 2) it demonstrates significant advantages in controlling Z-selectivity; 3) it achieves efficient deuteration, which was previously unattainable or at least not realized by earlier researchers; 4) it boasts the broadest and most comprehensive scope of alkyne substrates, with various challenging substrates, including diaryl alkynes and alkynyl esters, being well-tolerated. The authors have conducted an extensive investigation into the substrate scope, demonstrating the broad applicability of their method to various types of alkyne substrates. The high Z-selectivity and deuteration rate achieved are impressive and validate the effectiveness of the proposed reaction.

The reviewer believes that this work provides a valuable method for the synthesis of deuterated Z-alkenes, which may have broad applications in organic synthesis and drug design. The manuscript is well-organized and clearly written. With minor revisions to improve clarity and completeness, the manuscript is well-suited for publication in Nature Communications.

- 1) The authors may consider discussing potential limitations of their method, such as the scalability and cost-effectiveness of using cobalt salts and deuterium sources on an industrial scale.
- 2) The reaction should involve the oxidation of PPh₃ as a sacrificial reagent, with the final product most likely being triphenylphosphine oxide. It is proposed to use heavy-oxygen water [H₂(¹⁸O)] under standard conditions to verify the cycling of oxygen atoms.
- 3) For electrochemical experiments, conducting CV experiments is highly necessary. Furthermore, it should be noted whether the metal/ligand ratio has an impact on the reduction peaks. And how about the reduction peaks after the addition of a proton source or the alkyne substrate? Experimental details can be referred to ref. 57.
- 4) The author should also consider conducting kinetic experiments to aid in understanding the reaction mechanism. Is the kinetic order of the substrate, catalyst, and proton source related to the current? Will the magnitude of the current affect the reaction rate step? Is the reaction rate determining step different under different currents?
- 5) Based on points 2-4, it would be beneficial to provide a brief discussion on the mechanisms underlying this electrocatalytic semi-deuteration of alkynes.
- 6) Compounds S9 and S11 lack ¹⁹F-NMR information.

Reviewer #2

(Remarks to the Author)

This paper by X. Lu and Y. Fu describes a new molecular catalytic system for the electro-semideuteration of alkynes catalyzed by a bipyridine cobalt complex. This strategy makes it possible to use simple AcOD/D₂O as deuterium sources, with the help of triphenylphosphine as a sacrificial reductant, to selectively prepare and isolate a wide variety of deuterated Z-alkenes, including analogues of pharmaceutical molecules or other relevant natural products. The study is very well presented and shows in depth the wide applicability of the system as well as a scale-up, making it important reading for the electrocatalytic hydrogenation community. Consequently, I would recommend publication after the following points have been addressed:

- 1) The methodology described in the article does not allow access to a molecule such as calcifediol-d₃ (Figure 1, A). Isn't

there a more relevant example?

2) An important reference is missing which describes a very similar method for electrocatalytic deuteration of alkenes [J. Am. Chem. Soc. 2022, 144, 17783].

3) It's also worth mentioning that Kaeffer's group (Figure 1, B) demonstrated an isotopic kinetic effect using BzOD instead of BzOH [J. Am. Chem. Soc. 2023, 145, 17103] which shows that electrocatalytic deuteration is also possible in their case with a similar deuterium source, although it has not been explored further.

4) The use of PPh₃ stoichiometrically is a major drawback. Have the authors tested other sacrificial reductants such as diisopropylethylamine or triethanolamine (probably more reactive than the triethylamine already tested)?

5) In my opinion, it would be important to verify the homogeneous nature of the catalysis by carrying out a non-rinse test, for instance (which consists of restarting a catalytic run without the cobalt complex but with the same working electrode and without rinsing it [ACS Catal. 2016, 6, 3727]).

6) Finally, a question remains as to the true source of deuterium atoms (AcOD or D₂O) during the reaction, which is tricky because scrambling is likely possible. I think it would be interesting to discuss this further. I imagine that if we use the combination AcOD and H₂O, the deuterium incorporation drops drastically due to scrambling, but what about an AcOH/D₂O mixture? Perhaps the incorporation yields would remain very high in this case? Then the source of deuterium would be D₂O alone, which is even more elegant.

Regarding this question, perhaps the authors could also try replacing water with a substance less prone to scrambling with AcOD (but maintaining good hydrogen-bonding properties)? Like ethanol or tert-butanol, perhaps?

Also a few minor points:

7) In Figure 2, would it be relevant to specify the concentration (in 1 for example) and what -/- means (0 or not determined)?

8) It would be clearer to specify anode and cathode in the following sentence "Zn(+) and Mg(+) electrodes were employed to replace the graphite electrode (entries 14-15), and it was found that the Mg(+) electrode performed better. Replacing the graphite electrode with a Ni Foam(-) electrode yielded unsatisfactory results"

Version 1:

Reviewer comments:

Reviewer #1

(Remarks to the Author)

The authors have fully addressed my comments. Publication as it is is recommended. Congratulations.

Reviewer #2

(Remarks to the Author)

The authors have done an admirable job on all the points raised by the reviewers. I have only one final point to raise before recommending the publication:

With regard to the proposed non-rinse test, it involves testing the catalytic properties of an electrode that has already been used (at least one catalytic cycle). We know that rinsing can remove metal particles adsorbed during the initial electrolysis, so it is recommended not to rinse. Has this test been carried out in the right way, i.e. a catalytic run with the cobalt complex and then a catalytic run without it, but with the same electrode unrinsed? Does the 4% yield indicated in the answer meet this test criterion? If so, I have no further comments and congratulate the authors on this work.

REVIEWER COMMENTS

Reviewer #1 (Remarks to the Author):

In this study, the authors report the efficient synthesis of deuterated alkenes through electrochemical reduction using inexpensive deuterated sources, namely deuterated water and deuterated acetic acid. Compared to previous literature, this work is highly novel and advantageous in several aspects as the following: 1) the reaction effectively inhibits over-reduction, thereby minimizing the generation of alkane by-products; 2) it demonstrates significant advantages in controlling Z-selectivity; 3) it achieves efficient deuteration, which was previously unattainable or at least not realized by earlier researchers; 4) it boasts the broadest and most comprehensive scope of alkyne substrates, with various challenging substrates, including diaryl alkynes and alkynyl esters, being well-tolerated. The authors have conducted an extensive investigation into the substrate scope, demonstrating the broad applicability of their method to various types of alkyne substrates. The high Z-selectivity and deuteration rate achieved are impressive and validate the effectiveness of the proposed reaction.

The reviewer believes that this work provides a valuable method for the synthesis of deuterated Z-alkenes, which may have broad applications in organic synthesis and drug design. The manuscript is well-organized and clearly written. With minor revisions to improve clarity and completeness, the manuscript is well-suited for publication in Nature Communications.

- 1) The authors may consider discussing potential limitations of their method, such as the scalability and cost-effectiveness of using cobalt salts and deuterium sources on an industrial scale.
- 2) The reaction should involve the oxidation of PPh₃ as a sacrificial reagent, with the final product most likely being triphenylphosphine oxide. It is proposed to use heavy-oxygen water [H₂(¹⁸O)] under standard conditions to verify the cycling of oxygen atoms.
- 3) For electrochemical experiments, conducting CV experiments is highly necessary. Furthermore, it should be noted whether the metal/ligand ratio has an impact on the reduction peaks. And how about the reduction peaks after the addition of a proton source or the alkyne substrate? Experimental details can be referred to ref. 57.
- 4) The author should also consider conducting kinetic experiments to aid in understanding the reaction mechanism. Is the kinetic order of the substrate, catalyst, and proton source related to the current? Will the magnitude of the current affect the reaction rate step? Is the reaction rate determining step different under different currents?
- 5) Based on points 2-4, it would be beneficial to provide a brief discussion on the mechanisms underlying this electrocatalytic semi-deuteration of alkynes.
- 6) Compounds S9 and S11 lack ¹⁹F-NMR information.

Reviewer #2 (Remarks to the Author):

This paper by X. Lu and Y. Fu describes a new molecular catalytic system for the electro-semideuteration of alkynes catalyzed by a bipyridine cobalt complex. This strategy makes it possible to use simple AcOD/D₂O as deuterium sources, with the help of triphenylphosphine as a sacrificial reductant, to selectively prepare and isolate a wide variety of deuterated Z-alkenes, including analogues of pharmaceutical molecules or other relevant natural products. The study is very well presented and shows in depth the wide applicability of the system as well as a scale-up, making it important reading for the electrocatalytic hydrogenation community. Consequently, I would recommend publication after the following points have been addressed:

- 1) The methodology described in the article does not allow access to a molecule such as calcifediol-d₃ (Figure 1, A). Isn't there a more relevant example?
- 2) An important reference is missing which describes a very similar method for electrocatalytic deuteration of alkenes [J. Am. Chem. Soc. 2022, 144, 17783].
- 3) It's also worth mentioning that Kaeffer's group (Figure 1, B) demonstrated an isotopic kinetic effect using BzOD instead of BzOH [J. Am. Chem. Soc. 2023, 145, 17103] which shows that electrocatalytic deuteration is also possible in their case with a similar deuterium source, although it has not been explored further.
- 4) The use of PPh₃ stoichiometrically is a major drawback. Have the authors tested other sacrificial reductants such as diisopropylethylamine or triethanolamine (probably more reactive than the triethylamine already tested)?
- 5) In my opinion, it would be important to verify the homogeneous nature of the catalysis by carrying out a non-rinse test, for instance (which consists of restarting a catalytic run without the cobalt complex but with the same working electrode and without rinsing it [ACS Catal. 2016, 6, 3727]).
- 6) Finally, a question remains as to the true source of deuterium atoms (AcOD or D₂O) during the reaction, which is tricky because scrambling is likely possible. I think it would be interesting to discuss this further. I imagine that if we use the combination AcOD and H₂O, the deuterium incorporation drops drastically due to scrambling, but what about an AcOH/D₂O mixture? Perhaps the incorporation yields would remain very high in this case? Then the source of deuterium would be D₂O alone, which is even more elegant.

Regarding this question, perhaps the authors could also try replacing water with a substance less prone to scrambling with AcOD (but maintaining good hydrogen-bonding properties)? Like ethanol or tert-butanol, perhaps?

Also a few minor points:

- 7) In Figure 2, would it be relevant to specify the concentration (in 1 for example) and what -/- means (0 or not determined)?
- 8) It would be clearer to specify anode and cathode in the following sentence "Zn(+) and Mg(+) electrodes were employed to replace the graphite electrode (entries 14-15), and it was found that the

Mg(+) electrode performed better. Replacing the graphite electrode with a Ni Foam(-) electrode yielded unsatisfactory results”

Point-by-point Response to Reviewers' Comments

Reviewer 1

Response: We thank the Reviewer 1 for positive comments on our reactions and kind support as “with minor revisions to improve clarity and completeness, the manuscript is well-suited for publication in Nature Communications”.

Question 1: The authors may consider discussing potential limitations of their method, such as the scalability and cost-effectiveness of using cobalt salts and deuterium sources on an industrial scale.

Response: We thank the Reviewer 1 for valuable feedback on our method. We have conducted a gram-scale reaction, and the reaction proceeds smoothly (0.941 g product, 85% yield, *Z:E* > 20:1, 92% D and 91% D). However, there is a substantial gap between our gram-scale reaction and industrial-scale production. In particular, industrial-scale production necessitates careful consideration of core issues such as the recycling of D₂O and the waste treatment.

Based on the suggestions of Reviewer 1, we have incorporated these discussions into the main text.

To address the concerns raised by Reviewer 1, we have devised a simple design. For an industrial-scale production, the reaction mixture may be extracted using D₂O and ethyl acetate. Following this, the aqueous phase (comprising D₂O, DMAc, electrolytes, etc.) undergoes distillation to recover D₂O and DMAc, while solid electrolytes precipitate at the bottom of the reactor for centralized disposal. The products, located in the ethyl acetate phase, can be separated through vacuum distillation.

We also conducted a comprehensive analysis of industrial-scale production, examining various aspects including equipment, raw materials, energy consumption, labor requirements, and waste treatment. The cost for the synthesis of *cis*-3-hexenyl acetate-*d*₂ is \$9848/kg.

Although our calculations may not be precise, they suggest that recycling D₂O is crucial if we want to control the costs of industrial-scale production.

The detailed calculation process is as follows.

1) Model reaction

We selected the synthesis of *cis*-3-hexenyl acetate-*d*₂ as the model reaction.

2) Scale, factory site, and equipment

The extensive simulation of a fine chemicals facility is projected to manufacture prototype chemicals at a daily production rate of 1 kilogram. Given an annual operational duration of 8,000 hours over 330 days, the projected annual yield is calculated as follows: 1 kg/day * 330 days = 330 kg.

The principal apparatus included in the production process encompasses electrochemical reactors, alongside a suite of essential laboratory-grade chemical equipment. The electrochemical equipment plays a pivotal role in the process. For a batch requiring a specific reaction solvent, the volume of the reactor is approximately 200 L (0.2

m³). Accurately estimating the cost of electrochemical equipment is challenging, hence we focus on the depreciation of critical components, such as electrodes. Currently, the average market price for ultra-high power graphite electrodes is established at \$4,000/ton (<https://www.fastmarkets.com/insights/graphite-market-faces-rising-costs-in-2022/>). Assuming a 0.2 m³ cell utilizes a 200-kilogram graphite electrode with a lifespan of three years for this component. This implies that a total of 990 kg (330 kg/year × 3 year) of the product is synthesized, with a corresponding consumption of 200 kg of graphite electrodes. As a result, the cost of equipment maintenance (attrition) is approximately \$0.81/kg (calculated as \$4,000/ton * 200 kg / 990 kg). Other costs, including those for plant facilities and storage, which are referenced from the literature [*J. Flow. Chem.* **1**, 74-89 (2011). <https://doi.org/10.1556/jfchem.2011.00015>]. Facility cost of the workshop are summarized.

Facility cost of the workshop

Capital costs				
Items	Quantity	Unit	Unit price	Total prices
Floor space required	30	m ²	\$100/m ² /year	\$3,000/year
Storage	30	m ²	\$100/m ² /year	\$3,000/year
Maintenance costs	-	-	-	\$500/year
Graphite electrode	66.7	kg	\$4/kg	\$266.8/year
Labour	10	person	\$1000/pers/month	\$120,000/year
Total	\$126,766/year			

3) Raw and auxiliary materials

Beyond the facility costs, the paramount expenditure is intimately connected to the procurement of raw materials. Given that this product remains within the realm of fine chemicals, obtaining bulk pricing is inherently challenging. The prices listed on reagent vendor websites (<https://www.sigmaaldrich.cn/CN/en/life-science/sigmaaldrich>) are typically tailored for laboratory-scale research purposes. Consequently, to more accurately estimate the costs associated with scale-up, we employ a comparable "index estimation method" to ascertain the expenses for large-scale fine chemical production, as delineated in Equation 1:

$$C_1 = C_2 \left(\frac{S_1}{S_2}\right)^n \quad \text{Equation 1}$$

C_1 —the price of large-scale production;

C_2 —the price of small-scale production;

S_1 —the current production scale;

S_2 —expanded production scale;

n —the scale index. When the scale is expanded by increasing the size of the device, $n = 0.6-0.7$; when the scale is expanded by increasing the number of devices and equipment, $n=0.8-1.0$.

Considering the specificity of fine chemicals, we have set the scale factor 'n' to 0.9. The scale expansion factor is 10 times. The base price was obtained from a reputable reagent supplier, Sigma-Aldrich.

Basic price of raw materials and accessories

Items	Classify	Basic prices (\$/kg)	Negotiated price (\$/kg)	Annual consumption (kg)	Total cost (\$/year)
3-Hexyn-1-ol	fine chemicals	1,950	245.5	253.8	62,308
Cobalt (II) bromide	fine chemicals	1,290	162.4	28.4	4,612
dtbpy (4,4'-Di- tert -butyl-2,2'-dipyridyl)	fine chemicals	20,305	2,556	69.5	177,642
AcOD (CH ₃ COOD)	fine chemicals	3,122	393	790.5	310,667
TBABF ₄	fine chemicals	3,490	439	852.6	374,318
PPh ₃	fine chemicals	103	13	2,716.5	35,315
DMAc	bulk chemicals	0.736	0.736	46,606	34,302
D ₂ O (used for reaction)	strategic supplies	2,900	365	5,178.5	1,890,153
D ₂ O (used for extraction)	strategic supplies	2,900	365	51,785	18,901,525
Silica gel	bulk chemicals	40	5	3,300	16,500
Ethyl acetate	bulk chemicals	0.764	0.764	46,700	35,680
Total	\$21,843,022				

4) Energy consumption

The energy consumption for fine chemicals is primarily electrical, as operations such as electrochemical reactors, heating, and distillation all rely on electric heating. The electrochemical process is engineered with an energy requirement of 10 kWh/kg. The reaction byproducts predominantly consist of DMAc and D₂O, both of which are earmarked for recycling. Given the significant disparity in boiling points between D₂O and DMAc, their separation is feasible using a standard laboratory batch distillation column, with an assumed yield of 90%. The electrical energy consumption for batch distillation is calculated based on the enthalpy of vaporization {576.7 kJ/kg for DMAc [*Fluid. Phase Equilib.* **507**, 112437 (2020), <https://doi.org/10.1016/j.fluid.2019.112437>] and 2076 kJ/kg for D₂O}. Reflux ratio is 2.0.

Energy consumption

Items	Classify	Quantity	Unit	Unit price	Total prices (\$/year)
Electrochemical	electricity	3,300	kwh	\$0.14/kwh	462
D ₂ O recovery	electricity	65,700	kwh	\$0.14/kwh	9,200
DMAc recovery	electricity	15,000	kwh	\$0.14/kwh	2,100
Total	electricity	84,000	kwh	\$0.14/kwh	11,762

Following the recovery of solvents via distillation, the associated expenses for solvents are diminished.

Solvent recovery

Items	Classify	Basic prices	Negotiated price	Annual	Total cost
-------	----------	--------------	------------------	--------	------------

		(\$/kg)	(\$/kg)	consumption (kg)	(\$/year)
DMAc	bulk chemicals	0.736	0.736	-46,606×0.9	-30,872
D ₂ O	strategic supplies	2,900	365	-56,963.5×0.9	-18,712,510
Total	-18,743,382				

5) Waste treatment

Waste treatment includes liquid waste and solid waste. The waste liquid mainly contains organic solvents for extraction and column chromatography, while the solid waste contains metal salts, ligands, silica gel, etc. It is roughly estimated that the waste liquid is about 20 tons/year, and the solid-liquid is about 8 tons/year [J. Ind. Ecol. 27, 362-375 (2023), <https://doi.org/10.1111/jiec.13362>].

Waste treatment

Classify	Basic prices (\$/ton)	Annual consumption (ton)	Total cost (\$/year)
spent liquor	417	20	8340
solid waste	417	8	3336
Total	417	28	11,676

6) Conclusion and explanation

In all, this facility consumes $126,766 + 21,843,022 + 11,762 - 18,743,382 + 11,676 = 3,249,844$ dollars per year and generate 330 kg of *cis*-3-hexenyl acetate-*d*₂. That is, the cost for the synthesis of *cis*-3-hexenyl acetate-*d*₂ is \$9848/kg.

Question 2: The reaction should involve the oxidation of PPh₃ as a sacrificial reagent, with the final product most likely being triphenylphosphine oxide. It is proposed to use heavy-oxygen water [H₂(¹⁸O)] under standard conditions to verify the cycling of oxygen atoms.

Response: Reviewer 1's judgment is accurate. PPh₃ acts as a sacrificial reagent and is converted into PPh₃O.

Following the suggestions of Reviewer 1, we used heavy-oxygen water (H₂¹⁸O) instead of D₂O under standard conditions, and successfully obtained PPh₃¹⁸O. High-resolution mass spectrometry (HRMS) results indicate that ¹⁸O atoms originate from H₂¹⁸O.

Question 3: For electrochemical experiments, conducting CV experiments is highly necessary.

Furthermore, it should be noted whether the metal/ligand ratio has an impact on the reduction peaks.

And how about the reduction peaks after the addition of a proton source or the alkyne substrate?

Experimental details can be referred to ref. 57.

Response: We greatly appreciate Reviewer 1's suggestions. We have conducted CV experiments and investigated whether the metal/ligand ratio has an impact on the reduction peaks. Additionally, we have also studied the changes in the reduction peaks after the addition of a proton source or the alkyne substrate.

Cyclic voltammograms were recorded using a Signal1000E potentiostat at room temperature in DMAc within a glovebox. TBABF₄ (0.1 M) was used as the supporting electrolyte, and a 3 mm glassy carbon electrode served as the working electrode. The counter electrode was a coiled Pt wire, and the reference electrode was a commercial Ag/AgCl electrode. All potentials were referenced against ferrocene. The scan rate was set to 50 mV/s, with a sampling interval of 2 mV.

We tested the anodic oxidation section. PPh₃ can be oxidized at the anode, whereas the oxidation potential of Br⁻ is lower. Therefore, Br⁻ serves as an electron mediator in the oxidation reaction of PPh₃.

We examined various combinations of catalyst and ligand at different ratios. In the absence of a ligand, CoBr₂ exhibits a single reduction peak at -1.67 V. Upon the addition of dtbpy, the potential of this reduction peak shifts to -1.61 V (presumably corresponding to the reduction potential of Co^{II}/Co^I), and a new reduction peak emerges at -2.16 V, which we speculate is the reduction peak of Co^I/Co⁰.

During the gradual increase of dtbpy, some shoulder peaks emerge and then disappear, demonstrating that multiple species of the cobalt catalyst exist when dtbpy coordination is unsaturated, but this does not affect the main reduction peaks.

The blank dtbpy sample only exhibits a significant reduction peak near -2.7 V, proving that the reduction peaks in the catalyst-ligand ratio (CV) plot do not involve the reduction peaks of the ligand.

Upon the addition of alkynes (red line) or alkynes and D₂O (blue line) to the cobalt-ligand complex, the peak current at -2.16 V decreases, suggesting that coordination may occur between the alkyne and cobalt to affect the reduction of Co^I to Co⁰.

When AcOD (purple line) is directly added to the cobalt-ligand complex, the reduction peak at -2.16 V increases significantly, the peak current at -1.61 V decreases. When alkynes, water, and AcOD (green line) are simultaneously added to the cobalt-ligand complex, a significant reduction peak at -2.16 V appears, the peak current at -1.61 V decreases. We considered that the generation of new cobalt species might be the formation of CoH species.

Question 4: The author should also consider conducting kinetic experiments to aid in understanding the reaction mechanism. Is the kinetic order of the substrate, catalyst, and proton source related to the

current? Will the magnitude of the current affect the reaction rate step? Is the reaction rate determining step different under different currents?

Response: We have conducted kinetic experiments in accordance with Reviewer 1's suggestions.

At 2.5 mA, the reaction rates of the alkyne, cobalt catalyst, and AcOD under different excess conditions are basically the same. The initial induction period rates for the excess cobalt catalyst and AcOD are significantly faster than those of the standard reaction. However, by shifting the time axis, it can be observed that the mid-reaction rates are essentially consistent, all exhibiting zero-order kinetic behavior.

At 7.5 mA, the reaction rates remain constant for different excesses of alkyne and AcOD, and a significant increase in reaction rate is observed upon increasing the amount of cobalt catalyst, demonstrating that the kinetic order of both alkyne and AcOD remains zero. Through Bures graphical analysis for the reaction rates at different catalyst concentrations at 7.5 mA, it can be determined that the reaction rate has a first-order kinetics with respect to the cobalt catalyst at 7.5 mA.

Under a small current (2.5 mA), the rate-determining step of the reaction may be the electron transfer process; whereas, under a large current (7.5 mA), the rate-determining step shifts away from the electron transfer process. Based on the above experimental results, we believe that the rate-determining step of the reaction depends on the matching relationship between the current magnitude and the catalyst loading.

We hypothesize that, at a current of 2.5 mA, if the catalyst loading is significantly reduced, electron transfer saturation may also occur, resulting in a correlation of the reaction rate to the catalyst loading. Under this hypothesis, we attempted to use a smaller 2 mol% catalyst loading at a current of 2.5 mA, and the results obtained validated our conjecture.

Question 5: Based on points 2–4, it would be beneficial to provide a brief discussion on the mechanisms underlying this electrocatalytic semi-deuteration of alkynes.

Response: We compiled the results of the mechanistic experiments and incorporated them into Figure 2 of the revised manuscript. Based on these results, we propose a possible reaction mechanism involving $\text{Co}^{\text{II}}\text{H}$ species, which originate from the protonation of Co^0 species. After the insertion of the $\text{Co}^{\text{II}}\text{H}$ species into the alkyne, it undergoes a deuteration step to produce the semi-deuteration products. Moreover, cyclic voltammetry experiments indicate that the system relies on the more acidic AcOD. If Reviewer 1 has any additional questions or suggestions regarding our speculation, please kindly point them out to help us improve our understanding.

Question 6: Compounds **S9** and **S11** lack ^{19}F -NMR information.

Response: Thanks for Reviewer 1's reminder. We have supplemented the missing ^{19}F -NMR information for compounds **S9** and **S11**.

Reviewer 2

Response: We thank the Reviewer 2 for positive comments on our reactions and kind support as “recommend publication after the following points have been addressed”.

Question 1: The methodology described in the article does not allow access to a molecule such as calcifediol- d_3 (Figure 1, A). Isn't there a more relevant example?

Response: We thank the Reviewer 2 for this suggestion. Deuterated fatty olefinic acids play a significant role in the study of fatty acid metabolism. By introducing isotopic labels, they assist scientists in tracing the conversion pathways and metabolic processes of fatty acids. The design of deuterated fatty alkenes typically involves the

substitution of deuterium atoms at specific positions, with *cis*-deuterated fatty alkenes being used to investigate how the unsaturation of fatty acids changes during metabolism. For instance, the isotopically labeled molecule arachidonic acid- d_8 has been employed to enhance the Raman resolution of similar compounds in biological samples. The isotopically shifted Raman peaks offer insights into the orientation of deuterated Raman tags and their surrounding environments. [*J. Raman Spectrosc.* **53**, 297-309 (2022), <https://analyticalsciencejournals.onlinelibrary.wiley.com/doi/10.1002/jrs.6279>] In Figure 1A, we have replaced the original calcifediol- d_3 with arachidonic acid- d_8 .

Question 2: An important reference is missing which describes a very similar method for electrocatalytic deuteration of alkenes [*J. Am. Chem. Soc.* **2022**, *144*, 17783].

Response: Lin, Abruña, and their collaborators utilized both analytical and synthetic methodologies to investigate a series of electronically distinct Co(salen) complexes. They gained both qualitative and quantitative insights into the electroreductive formation of Co^{III}-H and its subsequent reaction with alkenes. These findings enabled the establishment of a strategy for reductive M-H formation, thereby facilitating the hydrofunctionalization of alkenes. We acknowledge the reviewer's suggestions and have cited this important reference accordingly.

Question 3: It's also worth mentioning that Kaeffer's group (Figure 1, B) demonstrated an isotopic kinetic effect using BzOD instead of BzOH [*J. Am. Chem. Soc.* **2023**, *145*, 17103] which shows that electrocatalytic deuteration is also possible in their case with a similar deuterium source, although it has not been explored further.

Response: We greatly appreciate Reviewer 2's reminder. Kaeffer's group reported that the electrocatalytic semi-hydrogenation of alkynes by molecular nickel-bipyridine complexes proceeds without the mediation of a hydride intermediate. They unveiled a mechanism primarily involving a nickelacyclopropene resting state upon alkyne binding to a low-valent Ni⁰ species. A subsequent sequence of protonation and electron transfer steps, via Ni^{II} and Ni^I vinyl intermediates, leads to alkene release in an overall ECEC-type pattern, which is identified as the most favored pathway. In this study, they employed BzOH and BzOD to investigate the kinetic isotope effect (KIE), and these mechanistic experiments demonstrated that electrocatalytic deuteration is also feasible in their system. We acknowledge the reviewer's suggestions and have cited this important reference. In the revised manuscript, we have clearly indicated that Kaeffer's group has shown the potential of BzOD for achieving electrocatalytic semi-deuteration of alkynes.

Question 4: The use of PPh₃ stoichiometrically is a major drawback. Have the authors tested other sacrificial reductants such as diisopropylethylamine or triethanolamine (probably more reactive than the triethylamine already tested)?

Response: Reviewer 2 has pointed out a crucial aspect. The use of PPh₃ as a sacrificial reagent results in the generation of PPh₃O waste, which affects product separation and poses challenges for industrial-scale production. We screened various sacrificial anodes and reagents, including Et₃N, triethanolamine, DIPEA, and DABCO, as alternative sacrificial reagents. However, their yields or deuteration rates are far inferior to those of PPh₃. Ultimately, we decided to use PPh₃ as the sacrificial reagent. Fortunately, Weix and his collaborators have reported a convenient method for PPh₃O removal, which involves mixing ZnCl₂ with PPh₃O to form a PPh₃O–Zn complex for the removal of PPh₃O [*J. Org. Chem.* **82**, 9931-9936 (2017), <https://pubs.acs.org/doi/10.1021/acs.joc.7b00459>].

Entry	Variation	GC Yield (%)	D content (%)
1	none	92	92 / 91
2	Et ₃ N instead of PPh ₃	16	not determined / not determined
3	triethanolamine instead of PPh ₃	41	72 / 70
4	DIPEA instead of PPh ₃	33	88 / 87
5	DABCO instead of PPh ₃	44	93 / 93

Question 5: In my opinion, it would be important to verify the homogeneous nature of the catalysis by carrying out a non-rinse test, for instance (which consists of restarting a catalytic run without the cobalt complex but with the same working electrode and without rinsing it [*ACS Catal.* **2016**, *6*, 3727]).

Response: We are grateful to Reviewer 2 for pointing out an issue that we had previously overlooked, namely verifying the homogeneous nature of the catalysis. Following Reviewer 2's suggestions, we conducted a non-rinse test.

Whether or not to rinse the electrode has little impact on the reaction yield and deuteration rate when cobalt catalyst and ligand are present. However, in the absence of cobalt catalyst and ligand, almost no target product can be obtained even if the electrode is not rinsed.

Question 6: Finally, a question remains as to the true source of deuterium atoms (AcOD or D₂O) during the reaction, which is tricky because scrambling is likely possible. I think it would be interesting to discuss this further. I imagine that if we use the combination AcOD and H₂O, the deuterium incorporation drops drastically due to scrambling, but what about an AcOH/D₂O mixture? Perhaps the incorporation yields would remain very high in this case? Then the source of deuterium would be D₂O alone, which is even more elegant. Regarding this question, perhaps the authors could also try replacing water with a substance less prone to scrambling with AcOD (but maintaining good hydrogen-bonding properties)? Like ethanol or *tert*-butanol, perhaps?

Response: We share Reviewer 2's curiosity about the true source of deuterium atoms. More importantly, if some of the high-priced reagents can be replaced with non-deuterated alternatives, especially without the need for large amount D₂O, it would be a great help for our reaction to move towards industrial-scale production.

We examined various combinations of H₂O, D₂O, AcOH, and AcOD, and observed that the reaction yield was minimally affected, whereas the deuteration rate was significantly influenced. When the combination of D₂O and AcOH was used, the deuteration rate dropped to 74% and 73% due to H-D exchange. Conversely, with the combination of H₂O and AcOD, the introduction of a large amount of H⁺ in the solvent resulted in products containing almost no deuterium atoms.

Additionally, using AcOD in combination with mixed solvents such as DMAc/EtOH or DMAc/*t*BuOH yielded products with unsatisfactory deuteration rates.

To minimize the use of costly deuterated reagents or solvents, we made several other attempts. For instance, we tried halving the amount of D₂O used. Naturally, the deuteration rate of the reaction remained almost unaffected, but there was a slight decrease in yield.

Entry	Variation	GC Yield (%)	D content (%)
1	none	92	92 / 91
2	AcOH and DMAc/D ₂ O used	87	74 / 73
3	AcOD and DMAc/H ₂ O used	85	<5 / <5
4	AcOH and DMAc/H ₂ O used	86	<5 / <5
5	AcOD and DMAc/EtOH used	61	5 / 5
6	AcOD and DMAc/ t BuOH used	34	7 / 7
7	DMAc/D ₂ O (3.6 mL/0.2 mL) used	87	92 / 91

Regarding the true source of deuterium atoms, it is indeed difficult to discuss. We hope to make a comprehensive judgment with CV experiments. Upon the addition of alkynes (red line) or alkynes and D₂O (blue line) to the cobalt-ligand complex, the peak current at -2.16 V decreases. When AcOD (purple line) is directly added to the cobalt-ligand complex, the reduction peak at -2.16 V increases significantly, the peak current at -1.61 V decreases. When alkynes, water, and AcOD (green line) are simultaneously added to the cobalt-ligand complex, a significant reduction peak at -2.16 V appears, and the peak current at -1.61 V decreases. We considered that the generation of new cobalt species might be the formation of CoH species after the addition of AcOD. These results indicate that

the system relies on the more acidic AcOD, and deuterium atoms may originate from AcOD.

If Reviewer 2 has any additional questions or suggestions regarding our speculation, please kindly point them out to help us improve our understanding.

Question 7: In Figure 2, would it be relevant to specify the concentration (in 1 for example) and what -/- means (0 or not determined)?

Response: In Figure 2, the alkyne substrate was used at a quantity of 0.20 mmol and a concentration of 0.050 mol/L, with CoBr₂ at 0.0025 mol/L, dtbpy at 0.005 mol/L, AcOD at 0.250 mol/L, PPh₃ at 0.200 mol/L, and TBABF₄ at 0.050 mol/L.

Our mistake has caused inconvenience to the reviewers. We are very grateful to Reviewer 2 for pointing out our oversight. ‘-/-’ in D content indicates that it has not been determined. We tested the reaction yields using gas chromatography. For conditions where the GC yield was below 30%, we did not proceed with isolation, purification, or subsequent D content NMR spectroscopy testing. Therefore, the deuteration rates for these reaction conditions are not provided.

Question 8: It would be clearer to specify anode and cathode in the following sentence “Zn(+) and Mg(+) electrodes were employed to replace the graphite electrode (entries 14-15), and it was found that the Mg(+) electrode performed better. Replacing the graphite electrode with a Ni Foam(-) electrode yielded unsatisfactory results”

Response: We have revised the relevant sentences as suggested.

Zn(+) and Mg(+) electrodes were used to replace the graphite electrode as the anode, while the cathode remained the graphite electrode. It was found that the Mg(+) electrode performed better than Zn(+). Replacing the graphite cathode electrode with a Ni Foam(-) electrode, while keeping the anode as the graphite electrode, yielded unsatisfactory results.

REVIEWER COMMENTS

Reviewer #1 (Remarks to the Author):

The authors have fully addressed my comments. Publication as it is is recommended. Congratulations.

Reviewer #2 (Remarks to the Author):

The authors have done an admirable job on all the points raised by the reviewers. I have only one final point to raise before recommending the publication:

With regard to the proposed non-rinse test, it involves testing the catalytic properties of an electrode that has already been used (at least one catalytic cycle). We know that rinsing can remove metal particles adsorbed during the initial electrolysis, so it is recommended not to rinse. Has this test been carried out in the right way, i.e. a catalytic run with the cobalt complex and then a catalytic run without it, but with the same electrode unrinsed? Does the 4% yield indicated in the answer meet this test criterion? If so, I have no further comments and congratulate the authors on this work.

Point-by-point Response to Reviewers' Comments

Reviewer 1

Response: We thank Reviewer 1 for positive comments on our revision work and support for the publication of our manuscript.

Reviewer 2

Response: We thank the Reviewer 2 for positive comments on our revision work and kind support for the publication of our manuscript.

Question 1: With regard to the proposed non-rinse test, it involves testing the catalytic properties of an electrode that has already been used (at least one catalytic cycle). We know that rinsing can remove metal particles adsorbed during the initial electrolysis, so it is recommended not to rinse. Has this test been carried out in the right way, i.e. a catalytic run with the cobalt complex and then a catalytic run without it, but with the same electrode unrinsed? Does the 4% yield indicated in the answer meet this test criterion? If so, I have no further comments and congratulate the authors on this work.

Response: We empathize with Reviewer 2's concern. The following figure shows our experimental results. We need to explain a few of these results.

Non-rinse tests were conducted to examine the catalytic properties of an electrode that had already been used in at least one catalytic cycle.

Brand new electrodes were purchased from different store on www.1688.com. In the non-rinse tests, only the newly purchased electrodes with sufficient rinse by ethyl acetate and acetone can be considered as 'brand new electrodes'. However, electrodes that have undergone deep cleaning, including sanding with sandpaper and multiple washes with ethyl acetate and acetone, are also considered as 'clean electrodes' in other experiments. Non-rinse electrodes refer to electrodes that had already been used in at least one catalytic cycle and only rinsed with ethyl acetate drops to remove solvents on the surface, without deeper cleaning, including sanding with sandpaper and multiple washes with ethyl acetate and acetone. Due to the ability of the reaction solution to wet the electrode surface, the reaction solution not only contains homogeneous cobalt catalyst, but also directly contains some reaction products. If these attached reaction solutions are carried into the next cycle, it can lead to unpredictable and irreproducible errors. We dropped ethyl acetate on the electrodes to wash off the previous reaction solutions. This very gentle rinse is necessary.

In addition, there were indeed some deposits with luster on the surface of the electrodes. However, these deposits did not affect the reaction. In the absence of cobalt and ligand, almost no target product (only with 4% yield) can be obtained.

We agree with Reviewer 2's comments that rinsing can remove metal particles adsorbed during the initial electrolysis. However, we also believe that we have conducted non-rinse tests in a reasonable way.

Based on the experimental results yielding only 4% products, we tend to believe that the vast majority of the cobalt catalyst exhibits homogeneous characteristics, but with the possibility of tiny amounts of cobalt catalyst precipitating out of the reaction system in a heterogeneous morphology.